# Applying Chitin Enhanced Diafiltration Process (CEFP) in Removing Cobalt from Synthetic Wastewater

**DOI:** 10.3390/membranes12121194

**Published:** 2022-11-27

**Authors:** Noureddine Elboughdiri, Djamel Ghernaout, Aicha Gasmi, Muhammad Imran Khan, Badia Ghernaout

**Affiliations:** 1Chemical Engineering Department, College of Engineering, University of Ha’il, P.O. Box 2440, Ha’il 81441, Saudi Arabia; 2Chemical Engineering Process Department, National School of Engineers Gabes, University of Gabes, Street Omar Ibn El-Khattab, Gabes 6029, Tunisia; 3Chemical Engineering Department, Faculty of Engineering, University of Blida, P.O. Box 270, Blida 09000, Algeria; 4Laboratory of Engineering Processes and Industrial Systems, Chemical Engineering Department, National School of Engineers of Gabes, University of Gabes, Street Omar ibn El-Khattab, Gabes 6029, Tunisia; 5Research Institute of Sciences and Engineering, University of Sharjah, Sharjah 27272, United Arab Emirates; 6Mechanical Engineering Department, Amar Tlidji University of Laghouat, Laghouat 03000, Algeria

**Keywords:** chitin, polymer enhanced diafiltration, heavy metals, adsorption isotherm, adsorption kinetics, wastewater treatment

## Abstract

This research aims to study the removal of Cobalt (Co) using chitin. The optimum conditions for removing Co were ascertained through batch experiments. This study involves the determination of chitin metal-binding efficiency by using a polymer enhanced diafiltration setup that utilizes a membrane process (ultrafiltration) to keep the Chitin. The effects of several parameters on sorption like pH, the concentrations of chitin, and Co were examined. The best efficiency was reached if the setup was run at pH < 6.3 (i.e., chitin p*K*_a_). At acidic conditions and by employing 6 g/L of chitin, Co level (20 mg/L) was decreased at 95%. To further investigate the kinetics of sorption for each gram of chitin, equilibrium experiments were carried out. For 1–100 mM Co, the performed rheological measurements show that chitin was observed to be moderately shear thickening at relatively lower levels (4 and 6 g/L); further, it was moderately shear thinning at slightly more important levels (12 and 20 g/L). Some improvement of the raw polymer will be necessary to enhance sorption to a sustainable limit and make this scheme an economically viable process.

## 1. Introduction

For the last three decades, industrial wastewater, and especially metal-containing effluent, has emerged as a main ecological challenge. Around the globe, metalworking industries employing or generating various kinds of metals produce a growingly huge number of metal-carrying effluents [1,2,3,4,5,6].

To lessen the organic and inorganic tenor of tannery wastewaters, several physicochemical, electrochemical, and biological approaches have been used including coagulation/flocculation, chemical oxidation, membrane filtration, solvent extraction, adsorption, incineration, electrocoagulation, aerobic or anaerobic biological treatment, and by integrating several processes (e.g., integrating coagulation with microfiltration or electrodialysis). Related the adsorption-based techniques, numerous adsorbents like clays, activated carbon obtained from diverse biological and non-biological materials, and chitosan-based composites have been found efficient in proved to allow for efficient detaining metals from effluents [7,8].

Cobalt (Co) is one of the abundant toxic metals. As a heavy metal, it may be amalgamated with other elements such as iron, nickel, and copper in the solar atmosphere, planets, and meteorites, etc. Co is employed in producing corrosion resistant alloys, glass, ceramics, varnishes and paints, and metal plating; however, its first usage remains in producing lithium and metal-nickel hydride batteries and besides producing by electroplating and tanning in the textile industry. Therefore, the generated effluents in these industries are rich in Co and need treatment before being released into the environment. A few studies have been dedicated to extracting and removing Co from wastewater [6,7,8,9,10]. 

The physicochemical processes used for the removal of Co, such as ion-exchange and precipitation, are very expensive in treating on a larger scale and can generate metal-containing sludges that are not easy to dispose of. The biosorption process is considered an excellent substitute for physicochemical processes. Another reason to use biopolymers is that they are a cheap resource and widely available [11,12,13,14].

Chitin is also chosen as the biosorbent because it is cost-effective and convenient for retaining heavy metals like Co. The elevated chitosan’s affinity to heavy metals attracts additional attention in developing chitosan-based membranes for eliminating such metals from wastewater, benefitting from both their adsorptive and water filtration abilities. The literature review shows that Chitin is more protonated at low pH and at this pH, it is apt to retain anions through electrostatic attraction [15,16,17,18,19]. To the best of our knowledge, this is the first time in which adsorption of Co from synthetic wastewater is performed using Chitin. This study aims to demonstrate the employment of Chitin as a biosorbent for the retention of Cobalt cations utilizing chitin enhanced diafiltration process (CEFP). This system uses an ultrafiltration (UF) membrane. 

This strategy allows for the retention of the Chitin-metal complex and does not allow the unfixed metal to exceed across the filtrate.

The permeate analysis and the concentration of residual Cobalt determination help to interpret the binding ability of Chitin. The study also determined the optimal pH and polymer concentration using CEFP, the amount of ligand bound to the macromolecule using equilibrium dialysis experiments, and the kinetics of binding to estimate the metal uptake and metal-polymer interactions. 

## 2. Materials and Methods

### 2.1. Study of Dynamic Metal-Binding Using Chitin in CEFP

Powdered chitin, which has a degree of de-acetylation of 28% and 49% in dilute acetic acid and water, respectively [16], was weighed and solubilized in an acetic acid (CH_3_COOH) solution (0.05 M). chitin with a purity of 90% was purchased from Sigma Aldrich and used to make all chitin solutions (2 to 20 g/L). All other chemicals were purchased from Merck. Cobalt (II) chloride hexahydrate (CoCl_2_·6H_2_O) was used to make the standard Cobalt solutions employed in all tests. A stock solution (1 g/L) was produced by mixing CoCl_2_·6H_2_O in deionized water. Different standard solutions’ concentrations (10–100 mg/L) were produced by carefully diluting the 1 g/L solution. The pH of the prepared solution was regulated employing 1M HCl and 1M NaOH.

A volume (500 mL) of chitin in CH_3_COOH solution was left in the recipient. Two pumps were used: the first to provide the metal directly into the recipient, which contains chitin solution, and the second to provide the metal chitin complex into the UF membrane.

The metal feed flow rate was fixed at the exact identical rate at which the filtrate quits the setup, and the trans-membrane pressure (TMP) was fixed at 7–9 psi. Different samples of permeate were cumulated at periodic times, and to ascertain the presence of Co, we used the spectrophotometric assay. At regular intervals, the permeate flow rate was quantified. The control test was performed in the absence of chitin to ascertain fixation to the surface of the membrane, and the temperature was maintained at 25 °C [20,21,22].

Different concentrations of chitin varying from 2 to 20 g/L were tested in the CEFP. The initial level of Co in the feed was 20 mg/L, the pH was kept at 4, the TMP of about 6 psi, and the volume of permeate was evaluated and registered during the course of the run. After that, the concentrations of the metal in the permeate were determined using Atomic Absorption Spectrophotometer AA-7000 (Shimadzu, Tunisia), and the measurement of the acidity was done using a pH-meter SevenDirect SD20 (Mettler Toledo, Tunisia). 

### 2.2. Equilibrium Study of Co Binding Properties of Chitin

We dissolved Chitin in the acetic acid (0.05 M) to prepare a Chitin solution of 2 g/L (pH = 4), and prepared Co solutions using 1 g/L stock by diluting it to 0.1 g/L. Into the cell, 40 mL of the Chitin solution was introduced on the membrane dialysis side while 40 mL of the prepared cobalt solution was included in opposite side. At 25 °C, the whole dialysis unit was shaked at 250 revolutions per minute. After every 120 min, 100 µL) ample were withdrawn and Cobalt concentration was measured using the diphenyl carbazide method. A parallel control test was performed without Chitin, in which only acetic acid (0.05 M) was introduced into recovery cell.

The pH influence on metal retention was followed at pH 2, 3, 4, and 5. Solutions Chitin were produced by merging it into acetate solutions at different pH values. A 0.1 g/L Co solution was formulated, also pH was correspondingly regulated to 2, 3, 4, and 5. Further, measure volume of Chitin solution (50 mL) was introduced on one side of cell, and Co solution (50 mL) was introduced on opposite side. After attaining equilibrium, different samples (50 µL) were taken following 1440 min to determine the remaining concentration of the Co.

For the equilibrium study, the dialysis was done at changing levels of Co from 10 to 120 mg/L. The concentration of Chitin was fixed at 2 g/L, and the pH was adjusted to 4. Further, measured volume of (50 mL) solutions of metal solutions and Chitin were introduced on one side and recovery side respectively into the feed cell. From feed cell, the specimens. The metal was introduced into removed samples to estimate the removal percentage of Co. A control test was realized without Chitin, employing acetic acid solution (0.05 M) into the recovery cell. After that, the final Co levels were evaluated, and the equilibrium isotherm was established.

### 2.3. Rheological Studies of the Chitin-Co Complex

The viscosity of the chitin-Co complex was studied using a Bohlin CS (controlled stress) rheometer (Lund Sweden N 11: 02, Tunisia), by applying shear stress varying from 0.75 to 1.5 mPa. Different concentrations of chitin, varying from 4 to 20 g/L at pH = 4, were prepared to examine the impact of various chitin concentrations on the rheological behavior in the occurrence of Co [23,24,25]. 

## 3. Results and Discussion

### 3.1. Kinetic Study and Equilibrium Isotherm Analysis

Figure 1 shows the reduction in Co level into the feed cell when the equilibrium is at pH = 3 with time. The initial concentration is 115 mg/L of Co in the absence of chitin, the equilibrium level of 56 mg/L was achieved following 12 h of dialysis. Whereas in the occurrence of 2 g/L chitin in the recovery cell, the equilibrium level of 16 mg/L was obtained after 12 h. At pH = 4 and starting from a Co level of 126 mg/L of, the equilibrium concentrations are reached after 12 h: 60 mg/L in the absence of chitin and 14 mg/L in the occurrence of 2 g/L of chitin in the recovery cell.

From the curves plotted in Figure 1, it can be affirmed that chitin works like a sorbent for Co since the Co concentration in the feed cell lessens as the process proceeds. A response period of 10–12 h is needed to attain the equilibrium in all four tests. Further, Figure 2 shows the equilibrium isotherm for chitin at pH = 4 following 24 h. at 25 °C. Theoretically, the fixation sites on the sorbent become saturated (*q*_max_) with an elevation in the equilibrium free Co concentration because of the uptake increase. Therefore, higher concentrations of Co (i.e., 350 mg/L) were tested as shown in Figure 3. 

Figure 2 and Figure 3 show that *q*_max_ is much higher than the maximum experimental value and the reciprocal linear isotherm plot is employed to determine *q_max_* and *K_d_* values from the empirical results. Figure 1 shows a reduction in the final free level of Co in the occurrence of chitin. This fact illustrates that fairly strong binding of Cobalt occurs and provides an idea of the period, which is needed by the metal-polymer reaction to attain the equilibrium [26,27,28]. A residence period of around 12 h is required to obtain the equilibrium.

The metal is adsorbed from the bulk solution onto fixation sites on the sorbent. This phenomenon of adsorption conducts to a drop of free Co amount in solution and an augmentation of Co fixed on the sorbent till an equilibrium is obtained [29,30]. At the equilibrium, the metal is dispersed between the two solid and liquid phases; such dispersion is illustrated by the adsorption isotherms (Figure 2 and Figure 3). 

The greatest adsorption for the sorbent-sorbate was greater than the maximum adsorption that could be practically estimated, for this *q*_max_ could not be experimentally attained using the Langmuir equation and the linear reciprocal curve obtained from the adsorption isotherm curve (Figure 4). The parameters *K*_d_ and *q*_max_ will be calculated using Equation (1):(1)$$\frac{1}{q_{e}}=\frac{1}{q_{max}K_{d}C_{eq}}+\frac{1}{q_{max}}$$

Figure 4 depicts the linear reciprocal plot of 1/*q*_e_ versus 1/*C*_eq._ Considering the slope and intercept of the straight line, *q*_max_ and *K*_d_ could be determined. The value of *K*_d_ was determined to be around 155 mg/L.

After the experiment, initially pH was adjusted and re-tested. The uptake results (mg Co/g of chitin) as a function of pH (before and after the dialysis experiment) are given in Table 1. 

The best pH was determined to be 4, pursued by pH = 3. Also, the lowest removal efficiency was obtained at pH = 5. For this reason, all the experiments of the rheology investigations were realized at pH = 4. 

### 3.2. Dynamic Heavy Metal-Binding Investigations Employing Chitin in CEFP

#### 3.2.1. CEFP with a Feed Cobalt Concentration 

The CEFP was initially run with a 10 mg/L Co of feed concentration. During this run, the chitin concentration was fixed at 4 g/L in the recipient, the volume employed was 0.5 L, pH = 4, and the TMP was maintained at 7 psi. Figure 5 presents the obtained results.

#### 3.2.2. CEFP without Chitin

To verify the postulations, a control test was performed without chitin in the recipient employing the Co feed solution (20 mg/L). A volume of 0.5 L from the acetic acid solution (0.05 M) was placed in the recipient instead of chitin. The obtained results are presented in Figure 6. These results indicate the absence of binding to the membrane.

#### 3.2.3. CEFP with Chitin Concentration

To examine the influence of various chitin concentrations on the CEFP, several levels of chitin were used (2–6 g/L). Each experiment was run by maintaining 20 mg/L Cobalt as a feed concentration. Table 2 shows the flow rate of permeate, the TMP, and pH measurements during 2, 4, 5, and 6 g/L chitin runs. In all performed experiments, the permeation curve was not the same as obtained in the control test (Figure 6).

There is an indication that bonding might exist between the Cobalt ions and chitin. This indication originates from the distinction between the experimental and the control data. The concentration of Cobalt passing into the permeate decreased when the concentration of chitin was increased, and less than 1 mg/L Cobalt was able to pass into the permeate with 6 g/L chitin.

Figure 7 depicts the permeation curve for different concentrations of Chitin with a 20 mg/L Co feed concentration. The following parameters are maintained constant during these trials: the volume of Chitin solution into the recipient (500 mL, the pH (4), and the feed concentration of Cobalt (20 mg/L).

#### 3.2.4. CEFP Employing a 6 g/L Chitin Treating Bigger amount of Co solution

Since a concentration of 6 g/L Chitin showed better performance in the previous CEFP runs, the CEFP was tested at this concentration. In the former run, 1.7 L volume was used even if huge volumes were as well utilized into setup this time as juxtaposed to the previous run. In this experiment, about 3 L of Cobalt solution at 20 mg/L and pH = 4 were processed for 5 h. A permeate level of 1.6 mg/L Cobalt was reached when a 3 L volume was processed. The volume of Chitin and the TMP were maintained at 0.5 L and 7.5–8.0 psi, respectively. Figure 8 presents the CEFP test’s permeation curve with 6 g/L Chitin and dealing with bigger amounts of Cobalt (around 3 L).

#### 3.2.5. CEFP with 20 g/L Chitin at pH = 2.5

Such an experiment was performed to evaluate the impact of an ultra-elevated level of chitin on the CEFP and retention of Co. To solubilize elevated levels of chitin in the acetic acid, a lower pH (2.5) was necessary and, hence, kept. Figure 9 illustrates the details of the CEFP using 20 g/L chitin, and shows that a very low concentration was detected in the permeate. Also, the volume of permeate gathered was minimal, which signifies that the amount of processed Co solution was also small. Following 3 h, only 0.3 L was treated because of a decline in permeate flux for the elevated viscosity of the chitin solution.

#### 3.2.6. CEFP with Buffer Diafiltration

Buffer diafiltration analysis was completed to potentially cancel the leakage of low molecular weight portions of chitin into the permeate and through the membrane. For this purpose, 0.05 M acetate buffer was poured into the recipient; also, the CEFP was performed for 1 hr prior to starting the introduction of Co. The CEFP was run under identical circumstances to the former ones (i.e., 4 g/L chitin, 20 mg/L Co, and pH = 4). The comparison of the obtained results with those under the same conditions (i.e., pH = 4, the concentration of the chitin 4 g/L, Co feed concentration 20 mg/L) even if without buffer diafiltration showed the absence of a decline in the level of Co in the permeate (Figure 9).

### 3.3. Data Modeling of the CEFP

The Langmuir sorption model was selected to describe the empirical details using Equation (1). Depending on the results obtained for the CEFP setup, the kinetic model for biosorption can be elaborated by determining a global and partial mass balance (Equation (2)):(2)$$V\frac{dC}{dt}=FC_{0}-FC-XV\frac{dq}{dt}\;$$

*F:* the inlet flow rate (L/h), *V*: the reaction volume (L), *X*: level of the biomass into solution (g/L). 

From Equations (1) and (2), we obtained the next differential equation (Equation (3)):(3)$$\frac{dC}{dt}=\frac{C_{0}-C}{\tau \left[1+\frac{K_{s}Xq_{max}}{{\left(C+K_{s}\right)}^{2}}\right]}$$

*τ*: the residence time in the reactor (h),*q_max_:* Polymer’s maximum adsorption capacity (mg/g),*K_S_*: the dissociation constant (mg/L).

Using the next parameters: 6 g/L biosorbent, *q_max_* = 310 mg/g (asymptotic estimated value), *K_S_* = 153 mg/L (experimental value), *C*_0_ = 20 mg/L (the initial concentration of *C*_0_), and Equation (3), a projected theoretical concentration of *C*_0_ in the effluent was determined (Figure 10). There is a slight inconsistency between the experimental data and the predicted values (Figure 10).

In Figure 10, the green triangles show the empirical details without Chitin, and the violet stars depict the determined levels of Cobalt with 6 g/L Chitin into the effluent. The calculated levels are directly determined by utilizing Equation (3). The empirical details attained for CEFP with 6 g/L Chitin are denoted by blue squares. All experiments are performed by using a feed with measured concentration of cobalt (20 mg/L).

Several authors [24,25,26,27,28,29] investigated the removal efficiency of divalent cations like Cu^2+^, Zn^2+^, and Ni^2+^ from dilute solutions using Chitin-enhanced UF. In this study, a similar principle of diafiltration was used and the obtained results show that Chitin in a dynamic system can bind Co, and elevated holding degrees were obtained in the CEFP trials. When the test was realized in the lack of Chitin, it was noted that no fixation to the CEFP membrane happened. In the presence of Chitin, the permeation curve has nearly a flat slope. A significant variation into curve in the lack of Chitin, which illustrates that there is a fixation phenomenon between Chitin and Co (Figure 6 and Figure 7). 

The Chitin concentration’s influence on binding was then investigated, and the results show that an elevation in the concentration of Chitin leads to better retention in the CEFP. Nonetheless, when more volume is processed through the system, it can be noted that there is a regular augmentation in Co transporting into permeate. 

As shown in Figure 8, employing 20 g/L Chitin permitted at most a so small amount of Co solution to be treated. Further, the retention of Co was elevated as there was no detected Co into permeate. There was a significant decrease into permeate flux because of elevated viscosities. For effective total uptake and separation of the ions, the molecular mass (MM) of the polymer must be selected to ensure them. Also, it has been observed that elevated MMs increase process cost by reducing permeate flux. For this reason, it is absolutely vital to select an optimum concentration of polymer for CEFP [22,23,24,25].

When CEFP details are anticipated employing the Langmuir model, a small difference is observed between the experimental and the theoretical findings (Figure 9). This shows that a change must be made to the model so that it better represents the mechanism of the process. 

### 3.4. Rheological Studies of the Chitin-Co Complex

The viscosity of the chitin-Co complex was plotted and was used to study the shear-thinning or shear-thickening behavior of the complex [29,30,31]. When 5–80 mM Co was added at lower concentrations of chitin (4 and 6 g/L), the results depicted an increasing tendency with the rise in shear stress and the complex behaved slightly shear-thickening. Figure 11 6 g/L Chitin’s rheological behavior in the occurrence of 5–80 mM Co. Further, a control test was performed with only 6 g/L of Chitin in the absence of Co. Figure 12 illustrates the change in Chitin’s (4 g/L) rheological behavior of in the occurrence of 5–80 mM Co. The control test showed, without any metal in both cases (i.e., 4 and 6 g/L), nearly Newtonian behavior (viscosity did not vary considerably with an augmentation in shear stress). However, when 1–100 mM Co was added at higher concentrations of chitin (12 and 20 g/L), the viscosity depicted a decreasing tendency with the rise in shear stress and the complex behaved slightly shear-thinning.

Figure 13 and Figure 14 show Chitin’s (12 g/L) rheological behavior in the occurrence of 1–100 mM Co and Chitin’s (20 g/L) rheological behavior in the occurrence of Co (1–100 mM), respectively. The control test in each trial was performed without Co. The control test for 12 g/L was Newtonian, and showed a shear-thinning behavior at a higher concentration (i.e., 20 g/L).

The rheological properties of the chitin-Co complex were examined. Such a complex was submitted to different shear stresses and changes in viscosity. The obtained results show, at low levels of chitin, the complex was conducted as a small shear-thickening fluid (Figure 11 and Figure 12). At more important levels, the viscosity of the complex increased with the shear stress as presented in Figure 13 and Figure 14.

The obtained findings showed that, for low levels and low shear rates, chitin acts nearly Newtonian, with no apparent variation in viscosity. The behavior of chitin in the control test performed without the presence of Co was observed and presented in Figure 11 and Figure 13. It seems reasonable to conclude that the shear thickening occurs due to Co neutralization by chitin molecules and intermolecular interaction. A shear-thinning behavior, owing to the intramolecular interaction, is seen at higher concentrations of chitin [6,9,11]. 

## 4. Conclusions

This work presents, for the first time, the adsorption of Co from synthetic wastewater using the chitin enhanced diafiltration process (CEFP). The CEFP experiments show that chitin can be considered a suitable candidate for biosorption of Co, and the essential results of the current study are summarized as follow:The use of a higher concentration of chitin improved the uptake. It was noted that fewer than 1 mg/L Co into the permeate moved when 6 g/L chitin was employed. Nevertheless, an elevated concentration (20 g/L) conducted a decline in permeate flux. Therefore, it is crucial to select the most favorable level of chitin that does not affect the permeate flux.The pH strongly influences the uptake; the optimum uptake amount occurred at pH = 4.It seems from the shear thickening behavior that the mutual action of amine groups from multiple chitin molecules is the major cause for the neutralization of lower Co chitin concentrations. This type of behavior is absent at high chitin concentrations.

Future work can involve the study of the recovery of Co from the Chitin-Co complex, regeneration of the polymer, and the industrialization of the process.

## Figures and Tables

**Figure 1 membranes-12-01194-f001:**
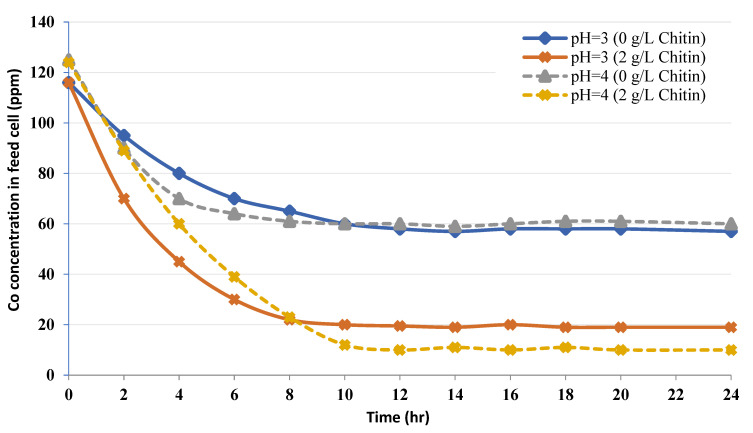
Kinetics of Co uptake by chitin at pH = 3 and pH = 4.

**Figure 2 membranes-12-01194-f002:**
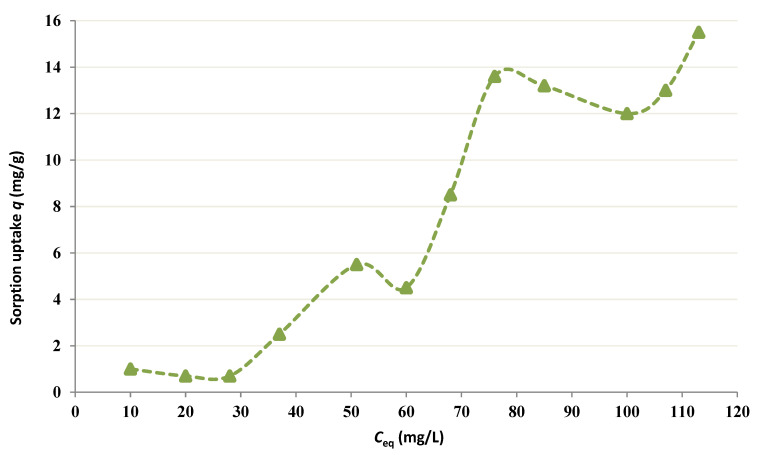
Adsorption isotherm for Co by chitin at pH = 4.

**Figure 3 membranes-12-01194-f003:**
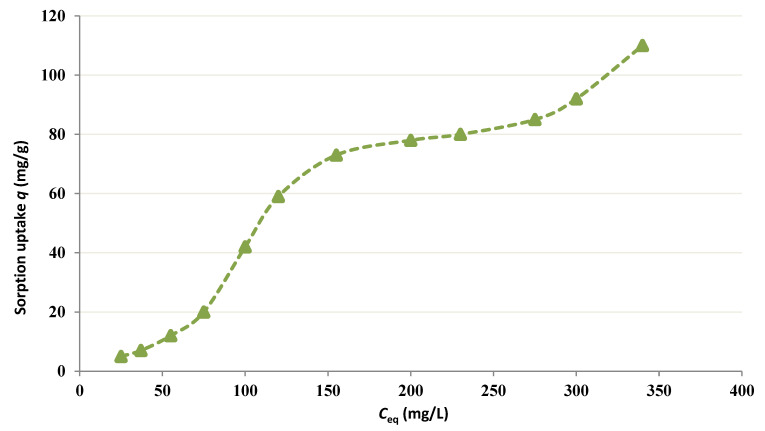
Adsorption isotherm for Co at higher equilibrium concentrations.

**Figure 4 membranes-12-01194-f004:**
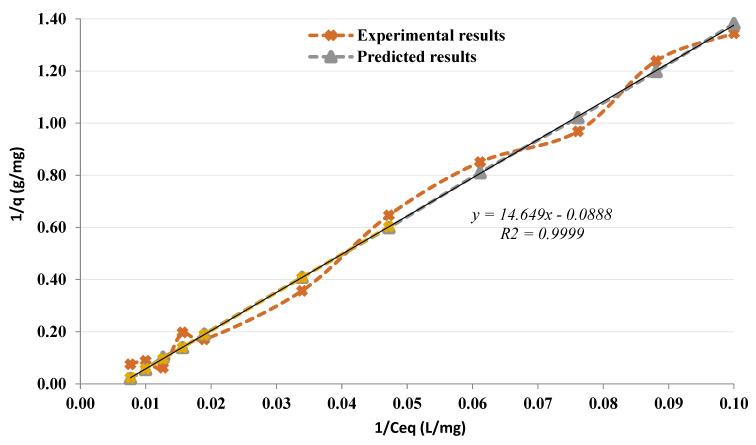
Reciprocal of the adsorption isotherm plot (1/*q*_e_ versus 1/*C*_eq_).

**Figure 5 membranes-12-01194-f005:**
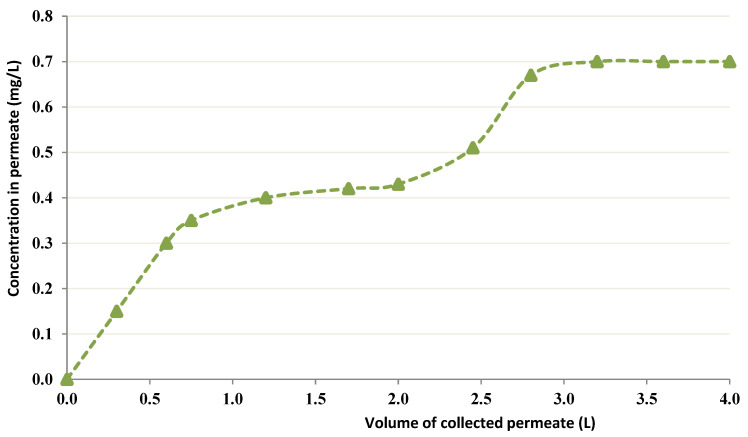
Permeation curve of Cobalt in the CEFP (Co feed concentration = 10 mg/L).

**Figure 6 membranes-12-01194-f006:**
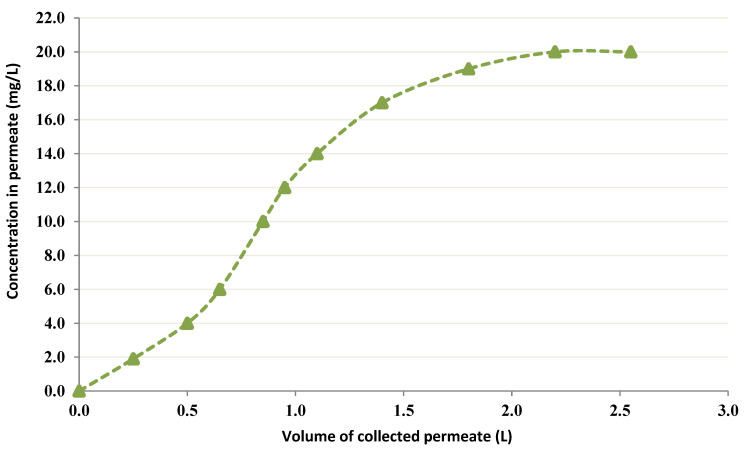
Permeation curve without chitin and with the Co feed solution (20 mg/L).

**Figure 7 membranes-12-01194-f007:**
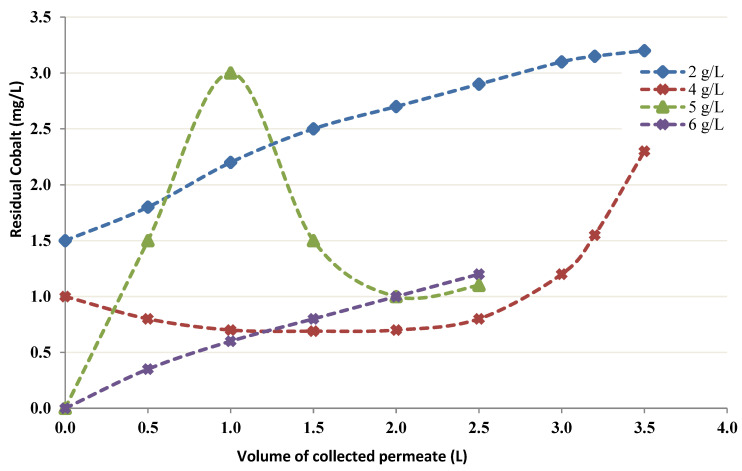
Permeation curve for various Chitin concentrations in CEFP.

**Figure 8 membranes-12-01194-f008:**
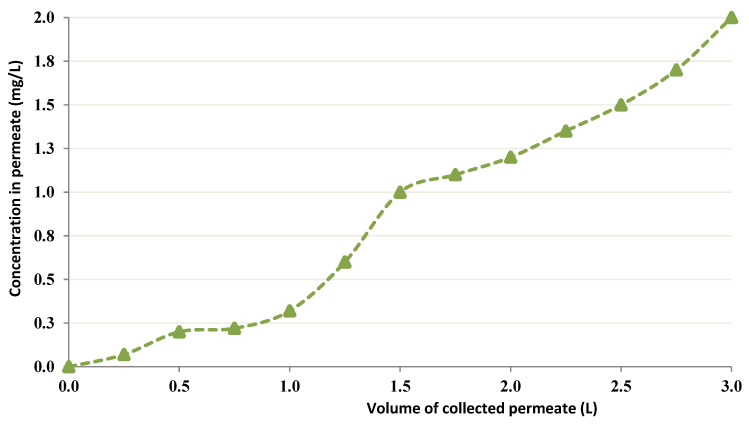
Permeation curve illustrating CEFP with 6 g/L chitin (volume of 20 mg/L Cobalt was 3 L, and pH = 4).

**Figure 9 membranes-12-01194-f009:**
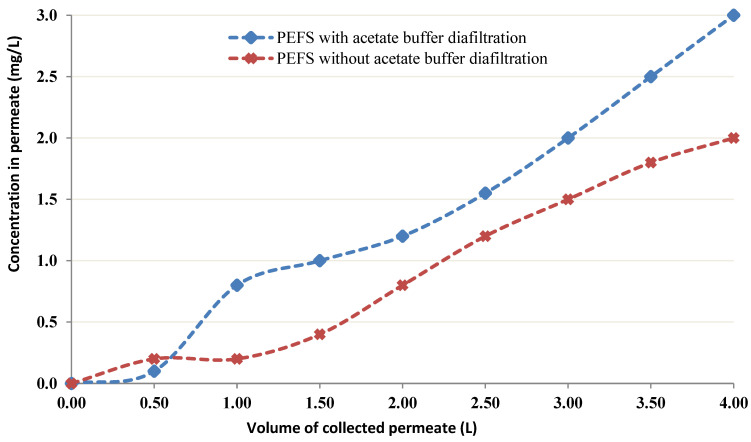
Permeation curves with 4 g/L chitin without and with buffer diafiltration (initial feed concentration of Co was 20 mg/L and pH = 4).

**Figure 10 membranes-12-01194-f010:**
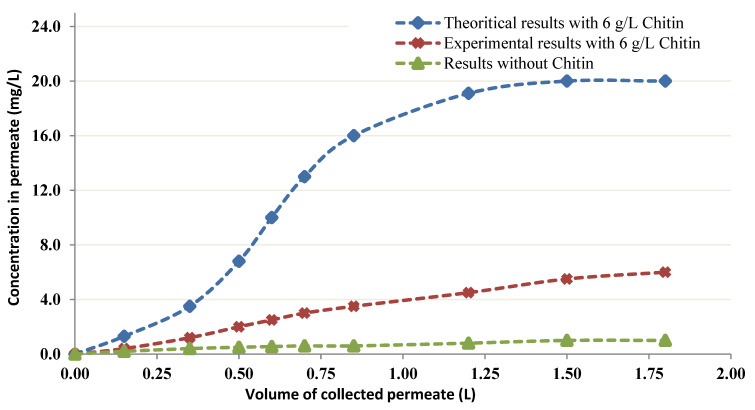
Permeation curve of chitin at 6 g/L with predicted curve at 6 g/L.

**Figure 11 membranes-12-01194-f011:**
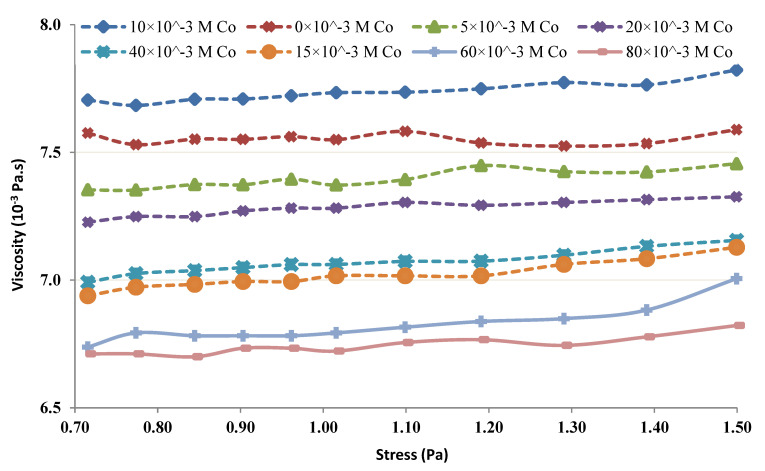
Rheological behavior of 6 g/L chitin.

**Figure 12 membranes-12-01194-f012:**
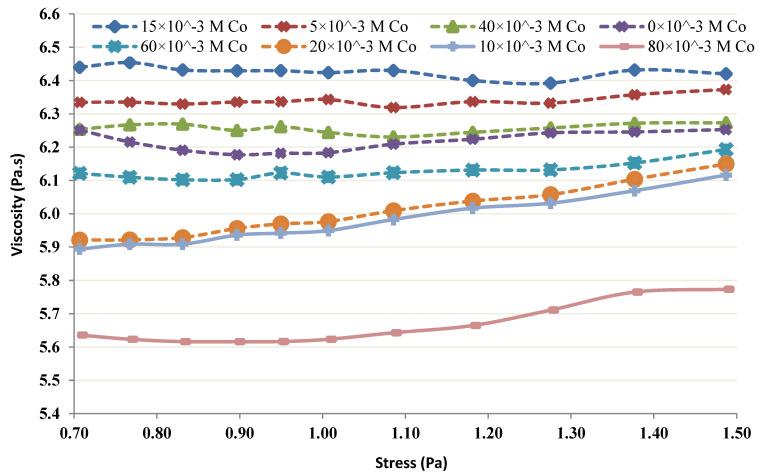
Rheological behavior of 4 g/L chitin.

**Figure 13 membranes-12-01194-f013:**
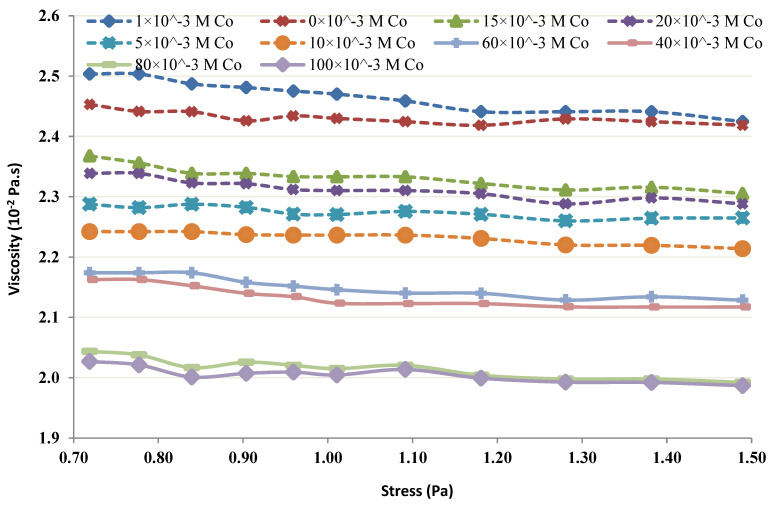
Rheological behavior of 12 g/L chitin.

**Figure 14 membranes-12-01194-f014:**
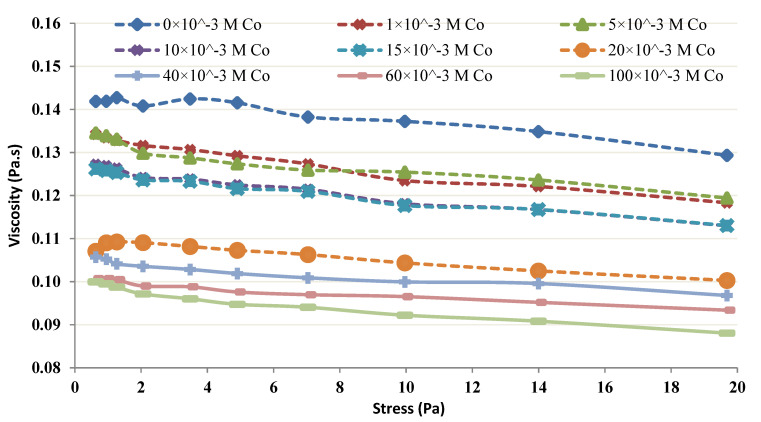
20 g/L Chitin’s Rheological behavior.

**Table 1 membranes-12-01194-t001:** Uptake by equilibrium dialysis dependence on pH (concentration of chitin = 2 g/L, initial concentration of Cobalt = 100 mg/L, temperature = 25 °C).

Initial pH	Uptake (mg/g)	Final pH
2	29	2.0
3	33	3.1
4	48	4.1
5	5	5.2

**Table 2 membranes-12-01194-t002:** CEFP with 20 mg/L Cobalt and various chitin concentrations effects on trans-membrane pressure (TMP), permeate flow rate, and pH.

Concentration of Chitin (g/L)	Volume of Permeate Collected (L)	TMP (psi)	Permeate Flow Rate (mL/min)	pH
2	0	6.74	29	4.1
	0.90	7.00	28	4.0
	1.70	7.25	28	4.1
	2.55	7.25	30	4.0
	3.45	7.25	30	4.0
4	0	7.0	22	4.1
	0.70	7.0	22	4.1
	1.40	7.25	22	4.1
	2.10	7.25	22	4.1
	2.75	7.25	22	4.0
	3.35	7.25	22	4.0
5	0	6.75	20	4.0
	0.60	6.75	20	4.1
	1.10	7.00	18	4.0
	1.50	7.00	15	4.0
	1.90	7.00	15	4.0
	2.25	7.00	15	4.0
6	0	7.5	13	4.0
	0.40	7.75	13	4.0
	0.75	7.75	13	4.0
	1.05	7.75	13	4.1
	1.40	7.5	13	4.1
	1.70	7.5	13	4.0

## Data Availability

Not applicable.

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
