# Peer review of "Applying Chitin Enhanced Diafiltration Process (CEFP) in Removing Cobalt from Synthetic Wastewater"

_membranes, 2022, doi:10.3390/membranes12121194_

Round 1
Reviewer 1 Report
The paper “Applying Chitin Enhanced Diafiltration Process (CEFP) in Re- 2 moving Cobalt from Synthetic Wastewater” by Elboughdiri et al. aims to prove the efficiency of Chitin in heavy metal ions removal from waste water.
The paper is well written and the effect of several parameters have been analyzed. Nevertheless, the main concern is related to the statement of authors who say they want to study the efficiency of Chitin in removing Cobalt anions (This study aims to demonstrate the employment of Chitin as a biosorbent for the retention of Cobalt anions utilizing CEFP). They add cobalt as CoCl2*6H2O, so they used Cobalt cations. Starting from that also the explanation of the observed rheological behaviour falls.
Please reformulate sentence lines 146-147: it is not so clear.
Authors should explain the reason they choose such a narrow range of stress to evaluate the viscosity trend
Author Response
Thank you to see the attached file

Reviewer 2 Report
- Abstract - Delete the first sentence or move it to another section
- The Introduction section should further highlight the novelty of their study.
- In the Introduction and Conclusions, the importance of research should be emphasized
- The choice of cobalt among other toxic metals must be justified.
- Line 58: “metal anion like Co” and Line 61 „of Cobalt anions” - should be corrected or clarified
- Figure 4 - unreadable line equation
Author Response
Thank you to see the attached file

Reviewer 3 Report
Reviewer 1: I read the manuscript entitled as "Applying Chitin Enhanced Diafiltration Process (CEFP) in Removing Cobalt from Synthetic Wastewater". In my opinion the presented manuscript can be published in "membranes" after minor revision. My comments are as follows:
1. The authors need to proofread the manuscript due to grammatical errors
2. Mention the novelty of work in the last paragraph of Introduction.
3. Line 73: the powdered Chitin was weighed and solubilized in an acetic acid (CH3COOH) solution (0.05 M), is chitin soluble in acetic acid?
4. Use error bars in figures 1,2,3,4,…..,14.
5. Add new references 2021,2022, I suggest to cite the following papers:
DOI: 10.3390/ma15155392, DOI: 10.1016/j.envres.2022.114294, DOI: 10.3390/w13182598
6. The conclusions should be rewritten
Author Response
Thank you to see the attached file

Round 2
Reviewer 1 Report
The paper has been significantly improved